# CK2 Inhibition and Antitumor Activity of 4,7-Dihydro-6-nitroazolo[1,5-a]pyrimidines

**DOI:** 10.3390/molecules27165239

**Published:** 2022-08-17

**Authors:** Daniil N. Lyapustin, Svetlana K. Kotovskaya, Ilya I. Butorin, Evgeny N. Ulomsky, Vladimir L. Rusinov, Denis A. Babkov, Alexander A. Pokhlebin, Alexander A. Spasov, Vsevolod V. Melekhin, Maria D. Tokhtueva, Anna V. Shcheglova, Oleg G. Makeev

**Affiliations:** 1Department of Organic and Biomolecular Chemistry, Ural Federal University, Mira St. 19, 620002 Ekaterinburg, Russia; 2Department of Pharmacology & Bioinformatics, Scientific Center for Innovative Drugs, Volgograd State Medical University, 400131 Volgograd, Russia; 3Department of Biology, Ural State Medical University, Repina 3, 620014 Yekaterinburg, Russia

**Keywords:** nitro compounds, Azolo[1,5-a]pyrimidines, CK2 inhibition, antitumor activity, multicomponent reaction

## Abstract

Today, cancer is one of the most widespread and dangerous human diseases with a high mortality rate. Nevertheless, the search and application of new low-toxic and effective drugs, combined with the timely diagnosis of diseases, makes it possible to cure most types of tumors at an early stage. In this work, the range of new polysubstituted 4,7-dihydro-6-nitroazolo[1,5-a]pyrimidines was extended. The structure of all the obtained compounds was confirmed by the data of ^1^H, ^13^C NMR spectroscopy, IR spectroscopy, and elemental analysis. These compounds were evaluated against human recombinant CK2 using the ADP-GloTM assay. In addition, the IC_50_ parameters were calculated based on the results of the MTT test against glioblastoma (A-172), embryonic rhabdomyosarcoma (Rd), osteosarcoma (Hos), and human embryonic kidney (Hek-293) cells. Compounds **5f**, **5h**, and **5k** showed a CK2 inhibitory activity close to the reference molecule (staurosporine). The most potential compound in the MTT test was **5m** with an IC_50_ from 13 to 27 µM. Thus, our results demonstrate that 4,7-dihydro-6-nitroazolo[1,5-a]pyrimidines are promising for further investigation of their antitumor properties.

## 1. Introduction

Cancer is one of the world′s leading causes of death, with an estimated number of 10 million deaths in 2020 [1]. However, many types of cancer are curable with early diagnosis and treatment. The establishment of alternative ways to treat tumor diseases allows for the use of new effective and low-toxic drugs in the early stages of the disease. One of the current trends is the inhibition of biological targets responsible for the growth, proliferation, and survival of tumor cells. From this point of view, type 2 casein kinase is a promising target for chemotherapy. The overexpression of casein kinase 2 (CK2) is closely associated with several cancers, including cancers of the head and neck, breast, kidney, lung, etc. [2,3,4,5,6,7,8,9], thus making CK2 a promising target for chemotherapy [10,11,12,13,14]. The ATP binding site of CK2 is smaller than that of most other kinases due to the presence of unique bulky residues, such as Val66 and Ile174, which create the prerequisites for the development of selective small molecule ATP-competitive inhibitors [15,16]. In the review article of CK2 and its inhibitors [17] by Iegre and colleagues, compounds of the azolo[1,5-a]pyrimidine series are noted as one of the most significant types of inhibitors over the past decade [18,19], along with azole derivatives(Figure 1) [20,21].

The antitumor activity of azolo[1,5-a]pyrimidines [22,23,24,25,26,27] has been related to the inhibition of cancer-associated kinases [28,29] (cyclin-dependent kinase 2 and phosphoinositide-3-kinase). However, recent Safari’s work demonstrates a positive trend in the cytotoxic effect of nitro-containing azolo[1,5-a]pyrimidines against human malignant melanoma cells (A375) and prostate cancer (PC3 cells, LNCaP cells) [30]. Examples of azolo[1,5-a]pyrimidines exhibiting antitumor activities are shown in Figure 2.

To continue our research on polysubstituted 6-nitroazolo[1,5-a]pyrimidines [31,32,33], we would like to present the synthesis of new compounds of this series, as well as their inhibitory activity against protein kinase CK2 and their cytotoxic effect against cultured tumor cells of human glioblastoma (A-172, ATCC CRL 1620), embryonic rhabdomyosarcoma (Rd, ATCC CRL 136), human osteosarcoma (Hos, ATCC CRL 1543), and human embryonic kidney (HEK-293).

## 2. Results and Discussion

### 2.1. Synthesis

In the present work, we studied compounds of the 4,7-dihydro-6-nitroazolo[1,5-a]pyrimidine **5a-o** and **6a-e** series. These compounds were obtained by a multicomponent reaction between aminoazoles **1,2**, 1-morpholino-2-nitroalkenes **3**, and aldehydes **4** (Figure 1). It was shown [31] that an initial reaction occurs between 1-morpholino-2-nitroalkenes **3** and aminoazoles **1,2**, followed by heterocyclization to products **5**, and the interaction of boron trifluoride etherate with the morpholinenitroalkene **3** leads to the formation of a corresponding alkyne and morpholinium tetrafluoroborate. The structure of all the obtained products **5,6** was confirmed by the data of ^1^H, ^13^C NMR spectroscopy, IR spectroscopy, and elemental analysis. The signals H-7 and C-7 are the characteristic for products **5**,**6** in the corresponding NMR spectra. It is interesting to note that in compounds **5a-d** obtained from 3-aminopyrazole **3a**, in the ^1^H spectra, the H-7 signal is in the region of 5.43–5.84 ppm, while in the ^13^C spectra, the characteristic C-7 signal is in the region of 34–40 ppm. In all other structures **5e-o, 6a-d**, these signals are shifted to a weaker region of the spectrum in the region of 6.44–6.94 and 55–60 ppm, respectively (see Appendix A). Apparently, the substituent and heteroatom in the azole ring affect the position of these signals.

### 2.2. CK2 Inhibition

Once in hand, target compounds were evaluated against human recombinant CK2 using the ADP-GloTM assay (Table 1). Initial screening at 50 μM revealed that compounds **5a**, **5c**, **5g**, **5m**, **5o**, **6a**, **6c**, and **6d** paradoxically enhance CK2 activity. Moderate inhibition was demonstrated by compounds **5f**, **5h**, **5k**, **5l**, and **6e**. Derivatives **5f**, **5h**, and **5k** were the most active inhibitors. One can notice that compounds bearing alkyl or alkylthio substituents at position C-2 and at position C-6 simultaneously tend to be more active, though the high structural similarity in this series does not allow us to define more comprehensive SAR. A dose–response study confirmed that compounds **5f**, **5h**, **5k**, and **5l** are micromolar CK2 inhibitors, while **6e** has a low potency (Table 2). A hill coefficient around (−1) indicates that lead compounds as well as staurosporine behave like classical inhibitors that bind to a single kinase site.

### 2.3. Antitumor Activity

The IC_50_ parameters were calculated based on the results of the MTT test (Table 3). The values are defined in the range from 13 µM to >650 µM. It should be noted that the study of the **5a**, **5b**, and **5c** cytotoxic effects was limited to exclude because of their low solubility.

Compounds **5j** and **6a** possessed the least pronounced cytotoxic effect on cells (in all cases IC_50_ > 0.4 mM), while the greatest decrease in the viability of tumor cells was noted with the addition of compounds **5m**, **5o**, **6c**, and **6d**. It is important to note that these azolopyrimidine compounds are characterized by a more pronounced suppression of the viability of tumor cells A-172, Rd, and Hos in comparison with the effect on human embryonic kidney cells Hek-293 (Figure 3 and Figure 4).

It was found that the IC_50_ for compounds **5m**, **5o**, **6c**, and **6d** in tumor cell lines studies, mostly, did not exceed 50 µM, whereas the cytotoxicity index on embryonic cells was higher than 169 µM (**5m**, **5o**, and **6c**). It should be noted that compounds **6d** containing a triazole fragment with a CF_3_-substituent in the structure had similar micromolar IC_50_ values for kidney cells and tumor cells. The least cytotoxic effect on non-tumor cells was determined for compound **6c** (227.50 µM). At the same time, our results indicate that compound **5m** may have the most pronounced antitumor properties.

Meanwhile, the mechanisms for the suppression of cultured cells growth remain unclear and require further research. We can assume, that experimental data of cytotoxic action are not fully explained by the effect on CK2. On the one hand, it was noted that azolo[1,5-a]pyrimidines **5k**, **5l**, and **6e** inhibit both the enzymatic activity of CK2 and the viability of tumor cells. On the other hand, compounds **5m**, **5o**, **6c**, and **6d** significantly inhibit the growth of neoplastic cells without affecting CK2. We can assume that the cytotoxic effect of the synthesized compounds may be due to the effect on other intracellular targets.

## 3. Materials and Methods

### 3.1. Chemical Experiment

Unless stated otherwise, all solvents and commercially available reactants/reagents were used as received. Non-commercial starting materials were prepared as described below or according to literature procedures. One-dimensional ^1^H and ^13^C NMR spectra, as well as two-dimensional ^1^H–^13^C HMBC experiments were acquired on a Bruker DRX-400 instrument (400 and 101 MHz, respectively) or a Bruker Avance NEO 600 instrument (600 and 151 MHz, respectively), equipped with a Prodigy broadband gradient cryoprobe, utilizing DMSO-*d*_6_ as solvent and TMS as internal standard. IR spectra were recorded on a Bruker Alpha FTIR spectrometer equipped with a ZnSe ATR accessory. Elemental analysis was performed on a PerkinElmer 2400 CHN analyzer. The reaction progress was controlled by TLC on Silufol UV-254 plates, eluent—EtOAc. Melting points were determined on a Stuart SMP3 apparatus at the heating rate of 7 °C/min. 1-Morpholino-2-nitroethylenes **3** were prepared according to a literature procedure [34].


*4,7-Dihydro-6-nitroazolo[1,5-a]pyrimidines 5,6; General procedure 1.*


A total of 3 Mmol (1.5 equiv., 0.37 mL) of BF_3_·Et_2_O was added to a suspension 2 mmol (1.0 equiv.) of corresponding aminoazole **1,2**, 2 mmol (1.0 equiv.) of nitroalkene 3, and 2 mmol (1.0 equiv.) of aldehyde **4** in 5 mL *n*-BuOH. The reaction mixture was heated on oil bath at 120 °C for 2 h. The resulting solution was cooled to room temperature and stirred 15 min. The obtained precipitate was filtered off, washed with 15 mL of i-PrOH. The precipitate was suspended in 50 mL of water, stirred for 5 min, filtered off again, and washed with 15 mL of water.


*4,7-Dihydro-6-nitroazolo[1,5-a]pyrimidines 5,6; General procedure 2.*


A total of 3 Mmol (1.5 equiv., 0.37 mL) of BF_3_·Et_2_O was added to a suspension 2 mmol (1.0 equiv.) of corresponding aminoazole **1,2**, 2 mmol (1.0 equiv.) of nitroalkene **3**, and 2 mmol (1.0 equiv.) of aldehyde **4** in 5 mL *n*-BuOH. The reaction mixture was heated on oil bath at 120 °C for 2 h. After heating, the resulting solution was concentrated under reduced pressure. To the residue, 20 mL of 2M Na_2_CO_3_ and 50 mL of water was added and stirred for 20 min. Solution was extracted twice with 20 mL of EtOAc. To a water phase, 15 mL of hexane was added and mixture was neutralized by diluted HCl to pH 7. Resulting mixture was stirred for 30 min, filtered off, and washed with water.

*6-Nitro-7-phenyl-4,7-dihydropyrazolo[1,5-a]pyrimidine* (**5a**). The reaction was performed according to the general procedure 1 employing 0.166 g (2 mmol, 1 equiv.) of 3-aminopyrazole **1a**, 0.316 g (2 mmol, 1 equiv.) of 1-morpholino-2-nitroethylene **3a**, and 0.20 mL (2 mmol, 1 equiv.) of benzaldehyde **4a**. The product was recrystallized from DMF. The substance was dried over P_2_O_5_ at 170 °C. Yellow solid. Yield 0.387 g (80%). mp 295–297 °C. IR Spectrum,ν cm^−1^: 1528, 1417 (NO_2_). ^1^H NMR (400 MHz, DMSO-*d*_6_): δ = 5.43 (1H, s, H-7); 7.10–7.30 (5H, m, Ph); 7.41 (1H, s, H-2); 8.36 (1H, d, H-5, *J* = 6.1 Hz); 10.88 (1H, d, NH, *J* = 6.4 Hz); 12.45 (1H, s, H-3). ^13^C {^1^H} NMR (101 MHz, DMSO-*d*_6_): δ = 38.2; 106.8; 124.3; 126.2; 126.5; 128.4; 127.3; 139.0; 144.2; 146.0. Anal. Calcd. for C_12_H_10_N_4_O_2_: C, 59.50; H, 4.16; N, 23.13. Found: C, 59.61; H, 4.20; N, 23.01.

*7-(Anthracen-9-yl)-5-ethyl-7-nitro-4,7-dihydropyrazolo[1,5-a]pyrimidine* (**5b**). The reaction was performed according to the general procedure 1 employing 0.166 g (2 mmol, 1 equiv.) of 3-aminopyrazole **1a**, 0.372 g (2 mmol, 1 equiv.) of 1-morpholino-2-nitroethylene **3a** and 0.412 g (2 mmol, 1 equiv.) of 9-anthracenecarbaldehyde **4b**. The product was recrystallized from *n*-BuOH. The substance was dried over P_2_O_5_ at 170 °C. Yellow solid. Yield 0.385 g (53%). mp 215 °C with decomp. IR Spectrum, ν cm^−1^: 1518, 1277 (NO_2_). ^1^H NMR (400 MHz, DMSO-*d*_6_): δ = 1.38 (3H, t, CH_2_-CH_3_, *J =* 7.3 Hz); 2.73–2.84, 3.16–3.26 (2H, m, CH_2_-CH_3_); 8.40 (1H, s, H-2); 6.88 (1H, s, H-7); 7.08 (1H, s, H-9ʹ); 7.26 (1H, m, H-7ʹ); 7.35 (1H, m, H-2ʹ); 7.52 (2H, m, H-6ʹ); 7.60 (2H, m, H-3ʹ); 7.98 (1H, d, H-5ʹ, *J =* 8.4 Hz); 8.06 (1H, d, H-4ʹ, *J =* 8.4 Hz); 8.11 (1H, d, H-8ʹ, *J =* 9.1 Hz); 8.40 (1H, s, H-2); 8.71 (1H, d, H-1ʹ, *J =* 9.1 Hz); ^13^C {^1^H} NMR (101 MHz, DMSO-*d*_6_): δ = 13.5; 28.0; 34.8; 106.7; 124.6 (2C); 124.9; 125.0; 125.2; 125.5; 126.8; 126.9; 127.1; 128.7; 129.4; 129.9; 130.4; 131.5; 132.2; 137.5; 145.1; 155.6. Anal. Calcd. for C_22_H_18_N_4_O_2_: C, 71.34; H, 4.90; N, 15.13. Found: C, 71.52 H, 4.72; N, 14.99.

*6-Nitro-5-methyl-7-(4ʹ-nitrophenyl)-4,7-dihydropyrazolo[1,5-a]pyrimidine* (**5c**). A total of 3 Mmol (1.5 equiv., 0.37 mL) of BF_3_·Et_2_O was added to a suspension of 0.166 g (2 mmol, 1 equiv.) of 3-aminopyrazole **1a** and 0.344 g (2 mmol, 1 equiv.) of 1-morpholino-2-nitropropylene **3b** in 5 mL *n*-BuOH. The reaction mixture was heated on oil bath at 80 °C for 15 min. After this, 0.302 g (2 mmol, 1 equiv.) of 4-nitrobenzaldehyde **4c** was added to the obtained solution. The reaction mixture was heated on oil bath at 120 °C for 2 h. The resulting solution was cooled to room temperature and stirred for 15 min. The obtained precipitate was filtered off, washed with 15 mL of *i*-PrOH. The precipitate was suspended in 50 mL of water, stirred for 5 min, filtered off again and washed with 15 mL of water. To the residue, 20 mL of 2 M Na_2_CO_3_ and 50 mL of water were added and stirred for 20 min. The solution was extracted twice with 20 mL of EtOAc. To the water phase, 15 mL of hexane was added, and the mixture was neutralized by diluted HCl to pH 7. The resulting mixture was stirred overnight, filtered off, and washed with water. Yellow solid. Yield 0.355 g (59%). mp 198 °C with decomp. IR Spectrum, ν cm^−1^: 1535, 1352 (NO_2_); 1508, 1268 (NO_2_). ^1^H NMR (600 MHz, DMSO-*d*_6_): δ = 2.66 (3H, s, C-5-CH_3_); 5.64 (1H, s, H-5); 7.45 (1H, s, H-2); 7.50 (2H, d, H-2ʹ, *J =* 8.3 Hz); 8.12 (2H, d, H-3ʹ, *J =* 8.3 Hz); 10.95 (1H, s, NH); 12.49 (1H, s, H-3). ^13^C {^1^H} NMR (151 MHz, DMSO-*d*_6_): δ = 22.0; 39.8; 105.6; 121.8; 123.8; 127.0; 127.6; 144.2; 145.9; 152.5; 154.1. Anal. Calcd. for C_13_H_11_N_5_O_4_: C, 51.83; H, 3.68; N, 23.25 Found: C, 51.89; H, 3.73; N, 23.19.

*6-Nitro-5-methyl-7-(thiophen-2ʹ-yl)-4,7-dihydropyrazolo[1,5-a]pyrimidine* (**5d**). The reaction was performed according to the general procedure 1 employing 0.166 g (2 mmol, 1 equiv.) of 3-aminopyrazole **1a**, 0.344 g (2 mmol, 1 equiv.) of 1-morpholino-2-nitropropylene **2b** and 0.184 mL (2 mmol, 1 equiv.) of thiophen-2-carbaldehyde **4f**. Pale green solid. Yield 0.278 (53%). mp 210 °C with decomp. IR Spectrum, ν, cm^−1^: 1511, 1256 (NO_2_). ^1^H NMR (400 MHz, DMSO-*d*_6_): δ = 2.58 (3H, s, C-5-CH_3_); 5.84 (1H, s, H-7); 6.80–6.82 (1H, m, H-3ʹ); 6.83–6.87 (1H, m, H-4ʹ); 7.23 (1H, d, H-5ʹ, *J =* 5.0 Hz); 7.56 (1H, s, H-2); 10.85 (1H, s, NH); 12.49 (1H, s, H-3). ^13^C {^1^H} NMR (101 MHz, DMSO-*d*_6_): δ = 21.8; 34.3; 106.2; 122.9; 123.1; 123.6; 126.5; 126.7; 144.5; 150.6; 151.1. Anal. Calcd. for C_11_H_10_N_4_O_2_S: C, 50.37; H, 3.84; N, 21.36. Found: C, 50.20; H, 3.99; N, 21.49.

*3-Etoxycarbonyl-5-ethyl-6-nitro-7-phenyl-4,7-dihydropyrazolo[1,5-a]pyrimidine* (**5e**). The reaction was performed according to the general procedure 1 employing 0.31 g (2 mmol, 1 equiv.) of 3-amino-4-etoxycarbonylpyrazole **1b**, 0.372 g (2 mmol, 1 equiv.) of 1-morpholino-2-nitrobutylene **3c** and 0.2 mL (2 mmol, 1 equiv.) of benzaldehyde **4a**. Orange yellow solid. Yield 0.424 g (62%). mp 127–129 °C. IR Spectrum, ν, cm^−1^: 1669 (C=O); 1584, 1289 (NO_2_). ^1^H NMR (600 MHz, DMSO-*d*_6_): δ = 1.32 (6H, t, CH_2_-CH_3_, *J* = 7.1 Hz); 3.19 (2H, q, C-5-CH_2_-CH_3_, *J* = 7.2 Hz); 4.18–4.38 (2H, m, C(O)-CH_2_-CH_3_); 6.55 (1H, s, H-7); 7.22–7.36 (5H, m, Ph); 7.66 (1H, s, H-2); 10.29 (1H, s, NH). ^13^C {^1^H} NMR (151 MHz, DMSO-*d*_6_): δ = 12.5; 14.3; 25.2; 59.4; 59.8; 97.6; 122.0; 127.1; 139.6; 128.4; 128.6; 137.4; 140.8; 152.7; 161.7. Anal. Calcd. for C_17_H_18_N_4_O_4_: C, 59.64; H, 5.30; N, 16.37. Found: C, 59.69; H, 5.32; N, 16.49.

*3-Etoxycarbonyl-5-ethyl-6-nitro-7-(4ʹ-nitrophenyl)-4,7-dihydropyrazolo[1,5-a]pyrimidine* (**5f**). The reaction was performed according to the general procedure 2 employing 0.31 g (2 mmol, 1 equiv.) of 3-amino-4-etoxycarbonylpyrazole **1b**, 0.372 g (2 mmol, 1 equiv.) of 1-morpholino-2-nitrobutylene **3c** and 0.302 g (2 mmol, 1 equiv.) of 4-nitrobenzaldehyde **4c**. Pale yellow solid. Yield 0.425 g (55%). mp 156–158 °C. IR Spectrum, ν, cm^−1^: 1720 (C=O); 1579, 1519, 1350, 1307 (NO_2_). ^1^H NMR (600 MHz, DMSO-*d*_6_): δ = 1.34 (6H, m, CH_2_-CH_3_); 3.23 (3H, q, C-5-CH_2_-CH_3_); 4.30 (2H, q, C(O)-CH_2_-CH_3_); 6.72 (1H, s, H-7); 7.59 (2H, d, H-2ʹ, *J* = 8.4 Hz); 7.69 (1H, s, H-2); 8.19 (2H, d, H-3ʹ, *J* = 8.4 Hz); 10.48 (1H, s., NH). ^13^C {^1^H} NMR (151 MHz, DMSO-*d*_6_): δ = 12.5; 14.3; 25.2; 58.8; 59.9; 97.9; 121.3; 123.9; 128.6; 141.2; 146.4; 147.4; 153.5; 161.6. Anal. Calcd. for C_17_H_17_N_5_O_6_: C, 52.95; H, 4.57; N, 18.10. Found: C, 52.71; H, 4.42; N, 18.08.

*3-Cyano-6-nitro-7-phenyl-4,7-dihydropyrazolo[1,5-a]pyrimidine* (**5g**). The reaction was performed according to the general procedure 1 employing 0.216 g (2 mmol, 1 equiv.) of 3-amino-3-cyanopyrazole **1c**, 0.316 g (2 mmol, 1 equiv.) of 1-morpholino-2-nitroethylene **3c** and 0.20 mL (2 mmol, 1 equiv.) of benzaldehyde **4a**. The product was recrystallized from MeOH. Yellow solid. Yield 0.219 g (41%). mp 262 °C with decomp. IR Spectrum, ν, cm^−1^: 2232 (CN); 1593, 1333 (NO_2_). ^1^H NMR (400 MHz, DMSO-*d*_6_): δ = 6.62 (1H, s, H-7); 7.20–7.47 (5H, m, Ph); 7.95 (1H, s, H-2); 8.47 (1H, s, H-5); 12.43 (1H, br.s.; NH). ^13^C {^1^H} NMR (101 MHz, DMSO-*d*_6_): δ = 59.6; 76.2; 112.6; 124.3; 127.5; 128.6; 128.7; 134.6; 138.8; 139.6; 143.1. Anal. Calcd. for C_13_H_9_N_5_O_2_: C, 58.43; H, 3.39; N, 26.21. Found: C, 58.49; H, 3.33; N, 26.29

*3-Cyano-5-ehtyl-6-Nitro-7-(4ʹ-nitrophenyl)-4,7-dihydropyrazolo[1,5-a]pyrimidine* (**5h**). The reaction was performed according to the general procedure 1 employing 0.216 g (2 mmol, 1 equiv.) of 3-amino-4-cyanopyrazole **1c**, 0.372 g (2 mmol, 1 equiv.) of 1-morpholino-2-nitrobutylene **3c** and 0.302 g (2 mmol, 1 equiv.) of 4-nitrobenzaldehyde **4c**. The substance was dried over P_2_O_5_ at 170 °C. Yellow solid. Yield 0.394 g (58%). mp 230 °C with decomp. IR Spectrum, ν, cm^−1^: 2231 (CN); 1584, 1350 (NO_2_); 1520, 1315 (NO_2_). ^1^H NMR (400 MHz, DMSO-*d*_6_): δ = 1.32 (3H, t, CH_2_-CH_3_; *J =* Hz); 2.90–3.15 (2H, m, CH_2_-CH_3_); 6.76 (1H, s, H-7); 7.63 (2H, d, H-2ʹ, *J =* 8.3 Hz); 7.95 (1H, s, H-2); 8.17 (2H, d, H-3ʹ, *J =* 8.3 Hz); 12.07 (1H, s, NH). ^13^C {^1^H} NMR (101 MHz, DMSO-*d*_6_): δ = 12.4; 25.5; 59.1; 75.9; 112.7; 121.0; 123.8; 128.8; 139.3; 143.5; 146.1; 147.5; 153.0. Anal. Calcd. for C_15_H_12_N_6_O_4_: C, 52.94; H, 3.55; N, 24.70. Found: C, 52.89; H, 3.47; N, 24.69.

*3-Cyano-5-ehtyl-6-Nitro-7-(4ʹ-metoxyphenyl)-4,7-dihydropyrazolo[1,5-a]pyrimidine* (**5i**).

The reaction was performed according to the general procedure 1 employing 0.216 g (2 mmol, 1 equiv.) of 3-aminopyrazole **1c**, 0.372 g (2 mmol, 1 equiv.) of 1-morpholino-2-nitrobutylene **3c** and 0.24 mL (2 mmol, 1 equiv.) of 4-metoxybenzaldehyde **4c**. The product was recrystallized from MeOH. Light-yellow solid. Yield 0.338 g (52%). mp 249 °C with decomp. IR Spectrum, ν, cm^−1^: 2229 (CN); 1577, 1307 (NO_2_). ^1^H NMR (400 MHz, DMSO-*d*_6_): δ = 1.10–1.48 (3H, m, CH_2_-CH_3_); 2.87–3.13 (2H, m, CH_2_-CH_3_); 3.71 (3H, s, O-CH_3_); 6.54 (1H, s, H-7); 6.79–6.94 (2H, m, H-3ʹ); 7.13–7.30 (2H, m, H-2ʹ); 7.92 (1H, s, H-2); 11.83 (1H, s, NH). ^13^C {^1^H} NMR (101 MHz, DMSO-*d*_6_): δ = 12.4; 25.4; 55.1; 59.2; 75.3; 112.9; 114.0; 122.0; 128.5; 131.5; 139.1; 143.1; 151.8; 159.3. Anal. Calcd. for C_16_H_15_N_5_O_3_: C, 59.07; H, 4.65; N, 21.53. Found: C, 59.17; H, 4.69; N, 21.44.

*3-Cyano-5-Ethyl-6-nitro-7-(3ʹ-methoxy-4ʹ-hydroxyphenyl)-4,7-dihydropyrazolo[1,5-a]pyrimidine* (**5j**). The reaction was performed according to the general procedure 1 employing 0.216 g (2 mmol, 1 equiv.) of 3-amino-4-cyanopyrazole **1c**, 0.372 g (2 mmol, 1 equiv.) of 1-morpholino-2-nitrobutylene **3c** and 0.304 g (2 mmol, 1 equiv.) of 3-metoxy-4-hydroxybenzaldehyde **4e**. Pale yellow solid. Yield 0.355 g (52%). mp 241 °C with decomp. IR Spectrum, ν, cm^−1^: 3181 (OH); 2234 (CN); 1579, 1306 (NO_2_). ^1^H NMR (400 MHz, DMSO-*d*_6_): δ = 1.32 (3H, t, CH_2_-CH_3_, *J =* 7.3 Hz); 2.90–3.13 (2H, m, CH_2_-CH_3_); 3.74 (1H, s, O-CH_3_); 6.50 (1H, S, H-7); 6.62–6.69 (1H, m, H-5ʹ); 6.72 (1H, d, H-6ʹ, *J =* 8.1 Hz); 7.93 (1H, s, H-2); 9.13 (1H, s, OH); 11.79 (1H, s, NH). ^13^C {^1^H} NMR (101 MHz, DMSO-*d*_6_): δ = 12.5; 25.5; 55.7; 59.5; 75.3; 111.8; 112.9; 115.5; 119.5; 121.9; 130.2; 143.0; 147.0; 147.4; 151.7. Anal. Calcd. for C_16_H_15_N_5_O_4_: C, 56.30; H, 4.43; N, 20.52. Found: C, 56.39; H, 4.36; N, 20.44.

*3-Cyano-5-ethyl-6-nitro-7-(thiophen-2-yl)-4,7-dihydropyrazolo[1,5-a]pyrimidine* (**5k**). The reaction was performed according to the general procedure 1 employing 0.216 g (2 mmol, 1 equiv.) of 3-amino-4-cyanopyrazole **1c**, 0.372 g (2 mmol, 1 equiv.) of 1-morpholino-2-nitrobutylene **3c** and 0.18 mL (2 mmol, 1 equiv.) of thiophene-2-carbaldehyde **4f**. Pale red solid. Yield 0.307 g (51%). mp 188–190 °C with decomp. IR Spectrum, ν, cm^−1^: 2232 (CN); 1576, 1300 (NO_2_). ^1^H NMR (400 MHz, DMSO-*d*_6_): δ = 1.30 (3H, t, CH_2_-CH_3_, *J =* 7.4 Hz); 2.84–3.14 (2H, m, CH_2_-CH_3_); 6.94 (1H, s, H-7); 6.89–6.98 (1H, s, H-3ʹ); 7.01–7.08 (1H, m, H-4ʹ); 7.41–7.49 (1H, d, H-5ʹ, *J =* 4.9 Hz); 8.00 (1H, s, H-2); 12.00 (1H, s, NH). ^13^C {^1^H} NMR (101 MHz, DMSO-*d*_6_): δ = 12.8; 25.9; 55.1; 76.1; 113.2; 122.1; 127.0; 127.1; 127.5; 139.6; 142.4; 143.8; 152.7. Anal. Calcd. for C_13_H_11_N_5_O_2_S: C, 51.82; H, 3.68; N, 23.24. Found: C, 51.69; H, 3.66; N, 23.36.

*3-Cyano-2-methylthio-6-nitro-7-phenyl-4,7-dihydropyrazolo[1,5-a]pyrimidine* (**5l**). The reaction was performed according to the general procedure 1 employing 0.308 g (2 mmol, 1 equiv.) of 3-amino-4-cyano-5-methylthiopyrazole **1d**, 0.316 g (2 mmol, 1 equiv.) of 1-morpholino-2-nitroethylene **3a** and 0.20 mL (2 mmol, 1 equiv.) of benzaldehyde **4a**. Yellow solid. Yield 0.319 g (51%). mp 219 °C with decomp. IR Spectrum, ν, cm^−1^: 2229 (CN); 1596, 1325 (NO_2_). ^1^H NMR (400 MHz, DMSO-*d*_6_): δ = 2.41 (3H, s, S-CH_3_); 6.56 (1H, s, H-7); 7.11–7.56 (5H, m, Ph); 8.45 (1H, s, H-2); 12.44 (1H, br. s, NH). ^13^C {^1^H} NMR (101 MHz, DMSO-*d*_6_): δ = 13.8; 59.6; 76.1; 111.9; 124.6; 127.6; 128.6; 128.8; 134.3; 138.5; 140.8; 150.8. Anal. Calcd. for C_14_H_11_N_5_O_2_S: C, 53.67; H, 3.54; N, 22.35. Found: C, 53.75; H, 3.61; N, 22.19.

*3-Cyano-5-methyl-2-methylthio-6-nitro-7-phenyl-4,7-dihydropyrazolo[1,5-a]pyrimidine* (**5m**). The reaction was performed according to the general procedure 1 employing 0.308 g (2 mmol, 1 equiv.) of 3-amino-4-cyano-5-methylthiopyrazole **1d**, 0.344 g (2 mmol, 1 equiv.) of 1-morpholino-2-nitropropylene **3b** and 0.20 mL (2 mmol, 1 equiv.) of benzaldehyde **4a**. Sand color solid. Yield 0.294 g (45%). mp 251–253 °C with decomp. IR Spectrum, ν, cm^−1^: 2229 (CN); 1575, 1306 (NO_2_). ^1^H NMR (400 MHz, DMSO-*d*_6_): δ = 2.42 (3H, s, S-CH_3_); 2.67 (3H, s, C-5-CH_3_); 6.53 (1H, s, H-7); 7.27–7.47 (5H, m, Ph); 11.90 (1H, s, NH). ^13^C {^1^H} NMR (101 MHz, DMSO-*d*_6_): δ = 13.8; 19.6; 59.8; 75.3; 112.1; 122.6; 127.4 (2C); 128.6; 139.0; 140.4; 147.4; 150.9. Anal. Calcd. for C_15_H_13_N_5_O_2_S: C, 55.04; H, 4.00; N, 21.39. Found: C, 55.00; H, 4.04; N, 21.50.

*5-Ehtyl-3-cyano-2-methylthio-6-nitro-7-(4ʹ-nitrophenyl)-4,7-dihydropyrazolo[1,5-a]pyrimidine* (**5n**). The reaction was performed according to the general procedure 1 employing 0.308 g (2 mmol, 1 equiv.) of 3-amino-4-cyano-5-methylthiopyrazole **1d**, 0.376 g (2 mmol, 1 equiv.) of 1-morpholino-2-nitrobutylene **3c** and 0.302 g (2 mmol, 1 equiv.) of 4-nitrobenzaldehyde **4c**. The substance was dried over P_2_O_5_ at 170 °C. Yellow solid. Yield 0.363 g (47%). mp 220 °C with decomp. IR Spectrum, ν, cm^−1^: 2236 (CN); 1580, 1331 (NO_2_); 1524, 1350 (NO_2_). ^1^H NMR (400 MHz, DMSO-*d*_6_): δ = 1.30 (3H, t, CH_2_-CH_3_, *J* = 7.4 Hz); 2.40 (3H, s, S-CH_3_); 2.90–3.10 (2H, m, CH_2_-CH_3_); 6.71 (1H, s, H-7); 7.65 (2H, d, H-2ʹ, *J =* 8.3 Hz); 8.19 (2H, d, H-3ʹ, *J =* 8.3 Hz); 12.06 (1H, br.s.; NH). ^13^C {^1^H} NMR (101 MHz, DMSO-*d*_6_): δ = 12.3; 13.7; 59.0; 75.7; 112.0; 121.4; 129.9; 128.9; 140.6; 145.8; 147.5; 151.4; 152.7. Anal. Calcd. for C_16_H_14_N_6_O_4_S: C, 49.74; H, 3.65; N, 21.75. Found: C, 49.69; H, 3.60; N, 21.81.

*3-Cyano-2-methylthio-6-nitro-7-(4ʹ-nitrophenyl)-4,7-dihydropyrazolo[1,5-a]pyrimidine* (**5o**). The reaction was performed according to the general procedure 2 employing 0.308 g (2 mmol, 1 equiv.) of 3-amino-4-cyano-5-methylthiopyrazole **1d**, 0.372 g (2 mmol, 1 equiv.) of 1-morpholino-2-nitrobutylene **3c** and 0.243 mL (2 mmol, 1 equiv.) of 4-metoxybenzaldehyde **4d**. Yellow solid. Yield 0.378 g (51%). mp 183 °C with decomp. IR Spectrum, ν, cm^−1^: 2224 (CN); 1577, 1302 (NO_2_). ^1^H NMR (400 MHz, DMSO-*d*_6_): δ = 1.33 (3H, t, CH_2_-CH_3_, *J =* 7.3 Hz); 2.44 (1H, s, S-CH_3_); 3.01 (2H, q, CH_2_-CH_3_; *J =* 7.3 Hz); 3.75 (1H, s, O-CH_3_); 6.44 (1H, s, H-7); 6.84 (2H, d, H-3ʹ, *J =* 8.3 Hz); 7.19 (2H, d, H-2ʹ, *J =* 7.19 Hz); 11.68 (1H, s, NH). ^13^C {^1^H} NMR (101 MHz, DMSO-*d*_6_): δ = 12.8; 14.3; 25.9; 55.6; 59.7; 75.8; 112.6; 114.5; 122.8; 129.0; 131.7; 140.9; 151.3; 152.0; 159.9. Anal. Calcd. for C_17_H_17_N_5_O_3_S: C, 54.98; H, 4.61; N, 18.86. Found: C, 55.08; H, 4.59; N, 18.89.

*6-Nitro-7-phenyl-4,7-dihydro-1,2,4-triazolo[1,5-a]pyrimidine* (**6a**). The reaction was performed according to the general procedure 1 employing 0.168 g (2 mmol, 1 equiv.) of 3-amino-1,2,4-triazole **2a**, 0.316 g (2 mmol, 1 equiv.) of 1-morpholino-2-nitroethylene **3a** and 0.20 mL (2 mmol, 1 equiv.) of benzaldehyde **4a**. Yellow solid. Yield 0.253 g (52%). mp 269 °C with decomp. IR Spectrum, ν, cm^−1^: 1593, 1314 (NO_2_). ^1^H NMR (400 MHz, DMSO-*d*_6_): δ = 6.65 (1H, s, H-7); 7.15–7.60 (5H, m, Ph); 7.79 (1H, s, H-5); 8.54 (1H, s, H-2); 12.09 (1H, br. s., NH). ^13^C {^1^H} NMR (101 MHz, DMSO-*d*_6_): δ = 59.5; 123.8; 127.4; 128.6; 128.6; 136.5; 38.8; 145.7; 151.0. Anal. Calcd. for C_11_H_9_N_5_O_2_: C, 54.32; H, 3.73; N, 28.79. Found: C, 54.21; H, 3.79; N, 28.69.

*5-Methyl-2-methylthio-6-nitro-7-phenyl-4,7-dihydro-1,2,4-triazolo[1,5-a]pyrimidine* (**6b**). The reaction was performed according to the general procedure 1 employing 0.260 g (2 mmol, 1 equiv.) of 3-amino-5-methylthio-1,2,4-triazole **2b**, 0.344 g (2 mmol, 1 equiv.) of 1-morpholino-2-nitropropylene **3b** and 0.20 mL (2 mmol, 1 equiv.) of benzaldehyde **4a**. Pale yellow solid. Yield 0.327 g (54%). mp 274–276 °C. IR Spectrum, ν, cm^−1^: 1557, 1320 (NO_2_). ^1^H NMR (400 MHz, DMSO-*d*_6_): δ = 2.42(3H, s, S-CH_3_); 2.64 (3H, s, C-5-CH_3_); 6.56 (1H, s, H-7); 7.27–7.38 (5H, m, Ph); 11.90 (1H, s, NH). ^13^C {^1^H} NMR (101 MHz, DMSO-*d*_6_): δ = 13.5; 20.2; 59.8; 122.2; 127.4; 128.5; 128.6; 139.1; 146.2; 148.8; 160.0. Anal. Calcd. for C_13_H_13_N_5_O_2_S: C, 51.49; H, 4.41; N, 23.02. Found: C, 51.47; H, 4.32; N, 23.09.

*5-Methyl-2-methylthio-6-nitro-7-(4ʹ-nitrophenyl)-4,7-dihydro-1,2,4-triazolo[1,5-a]pyrimidine* (**6c**). The reaction was performed according to the general procedure 2 employing 0.260 g (2 mmol, 1 equiv.) of 3-amino-5-methylthio-1,2,4-triazole **2b**, 0.372 g (2 mmol, 1 equiv.) of 1-morpholino-2-nitrobutylene **2b** and 0.302 mL (2 mmol, 1 equiv.) of 4-nitrobenzaldehyde **4c**. Yellow solid. Yield 0.384 g (53%). mp 239–241 °C. IR Spectrum, ν, cm^−1^: 1581, 1555, 1345, 1303 (NO_2_). ^1^H NMR (400 MHz, DMSO-*d*_6_): δ = 1.34 (3H, t, CH_2_-CH_3_, *J =* 7.3 Hz); 2.44 (3H, s, S-CH_3_); 2.91–3.11 (2H, m, CH_2_-CH_3_); 6.71 (1H, s, H-7); 7.64 (2H, d, H-2ʹ, *J =* 8.5 Hz); 8.22 (2H, d, H-3ʹ, *J =* 8.6 Hz); 12.00 (1H, s., NH). ^13^C {^1^H} NMR (101 MHz, DMSO-*d*_6_): δ = 12.3; 13.5; 26.0; 59.0; 120.9; 123.84; 128.8; 146.0; 146.3; 147.5; 154.2; 160.5. Anal. Calcd. for C_14_H_14_N_6_O_4_S: C, 46.40; H, 3.89; N, 23.19. Found: C, 46.55; H, 3.77; N, 23.10.

*5-Methyl-6-nitro-7-phenyl-2-trifluoromethyl-4,7-dihydro-1,2,4-triazolo[1,5-a]pyrimidine* (**6d**).

To a suspension of 0.304 g (2 mmol, 1 equiv.) of 3-amino-5-trifluoromethyl-1,2,4-triazole **2c**, 0.344 g (2 mmol, 1 equiv.) of 1-morpholino-2-nitropropylene **3b** and 0.20 mL (2 mmol, 1 equiv.) of benzaldehyde **4a** in 5 mL *n*-BuOH 3 mmol (1.5 equiv., 0.37 mL) of BF_3_·Et_2_O was added. The reaction mixture was heated on oil bath at 120 °C for 2 h. The resulting solution was cooled to room temperature and evaporated. To residue 5 mL of *n*-heptane was added. The obtained suspension was stirred for 10 min, filtered off and washed with 20 mL of water. The product was recrystallized from *i*-PrOH-H_2_O 1/1. Yellow solid. Yield 0.273 g (42%); mp 233–235 °C. IR Spectrum, ν, cm^−1^: 1573, 1321 (NO_2_), 1133 (CF_3_). ^1^H NMR (400 MHz, DMSO-*d*_6_): δ = 2.66 (3H, s, C-5-CH_3_); 6.74 (1H, s, H-7); 7.31–7.45 (5H, m, Ph); 12.13 (1H, s, NH). ^13^C {^1^H} NMR (101 MHz, DMSO-*d*_6_): δ = 20.1; 60.4; 118.9 (q, *J =* 269.7 Hz); 122.5; 127.6; 128.7; 128.9; 138.4; 147.0; 148.8; 151.1 (q, *J =* 39.1 Hz). Anal. Calcd. for C_13_H_10_F_3_N_5_O_2_: C, 48.01; H, 3.10; N, 21.53. Found: C, 48.15; H, 3.24; N, 21.40.

*5-Methyl-6-nitro-7-(4ʹ-nitrophenyl)-2-trifluoromethyl-4,7-dihydro-1,2,4-triazolo[1,5-a]pyrimidine* (**6e**). 3 Mmol (1.5 equiv., 0.37 mL) of BF_3_·Et_2_O was added to a suspension of 0.304 g (2 mmol, 1 equiv.) of 3-amino-5-trifluoromethyl-1,2,4-triazole **2c**, 0.372 g (2 mmol, 1 equiv.) of 1-morpholino-2-nitrobutylene **3c** in 5 mL *n*-BuOH. The reaction mixture was heated on oil bath at 80 °C for 15 min. After this, 0.302 g (2 mmol, 1 equiv.) of 4-nitrobenzaldehyde **4c** was added to the obtained solution. The reaction mixture was heated on oil bath at 120 °C for 2 h. The resulting solution was cooled to room temperature and evaporated. To residue, 5×3 mL of *n*-heptane was added, and the obtained mixture was decanted. The same procedure was carried out with water. The crude oil was dissolved in 5 mL of *i*-PrOH, and the obtained solution was left overnight. The obtained suspension was filtered off and recrystallized from *i*-PrOH/H_2_O 1/1. Pale yellow solid. Yield 0.322 g (42%). mp 228 °C with decomp. IR Spectrum, ν, cm^−1^: 1573, 1309 (NO_2_); 1152 (CF_3_). ^1^H NMR (400 MHz, DMSO-*d*_6_): δ = 1.32 (3H, t, CH_2_-CH_3_, *J =* 7.4 Hz); 2.85–3.11 (2H, m, CH_2_-CH_3_); 6.93 (1H, s, H-7); 7.75 (2H, d, H-2ʹ, *J* = 8.3 Hz); 8.21 (2H, d, H-3ʹ, *J* = 8.3 Hz); 12.29 (1H, s, NH). ^13^C {^1^H} NMR (101 MHz, DMSO-*d*_6_): δ = 12.2; 26.0; 59.6; 118.81 (q, *J* = 269.9 Hz); 121.3; 124.0; 129.2; 145.1; 147.2; 147.8; 151.5 (q, *J* = 39.2 Hz); 154.3. Anal. Calcd. for C_14_H_11_F_3_N_6_O_4_: C, 43.76; H, 2.89; N, 21.87. Found: C, 43.69; H, 2.80; N, 21.98.

### 3.2. Biological Experiments

#### 3.2.1. CK2 Assay

Kinase activity was determined using the enzyme system CK2α1 (Promega V4482, Madison, WI, USA) and the ADP-Glo^TM^ kit (Promega V9101, Madison, WI, USA) in white 96-well plates (Nunc U96 Microwell 267350, Denmark). The reaction was carried out using 50 ng/well of N-GST labeled human recombinant CK2α1, 0.1 µg/µL bovine casein as a substrate, 10 μM ATP in 40 mM Tris buffer solution (pH 7.50) containing 20 mM MgCl_2_, 0.1 mg/mL of BSA, and 50 μM of DTT. Test compounds were added to 1.25% DMSO (final concentration 0.25%) and preincubated with kinase for 10 min. The reaction was carried out for 60 min at 25 °C in a thermostatically controlled PST-60HL shaker (Biosan. Beresfield, NSW, Latvia). ATP-dependent luminescence was measured at an integration time of 1000 ms using the Infinite M200 PRO microplate reader (Tecan. Austria). The ATP-competitive inhibitor Staurosporin (CAS 62996–74-1, Alfa Aesar J62837, 99+%) was used as a positive control. The experiments were performed in two parallels.

#### 3.2.2. Cytotoxicity Study

##### Cell Culture

The studies were carried out on cultured cells of human glioblastoma (A-172, ATCC CRL 1620) [35], human osteosarcoma (Hos, ATCC CRL 1543) [36,37,38], human embryonic rhabdomyosarcoma (Rd, ATCC CRL 136) [39], and human embryonic kidney 293 cells (Hek-293, ATCC CRL 1573) [40] obtained from the Shared research facility “Vertebrate cell culture collection” (Institute of Cytology RAS, Saint-Petersburg, Russia). The cells were cultured using DMEM / F-12 medium containing 10% fetal bovine serum at 37°C, 5% CO_2_, and 98% humidity. Subculturing was performed using 0.25% trypsin solution when the culture reached ≥90% confluency.

##### Viability Assessment

The compounds were dissolved in DMSO. The solutions were diluted with DMEM/F-12 culture medium with 10% fetal bovine serum to the studied concentrations: 8, 16, 32, 64, 128, 256, 512, and 1024 µM. In all cases, the concentration of DMSO in the final solution did not exceed 1%. Cisplatin (cPt) was used as a positive control.

Cells were seeded in 96-well plates at a concentration of 4 × 10^3^ cells per well. After 24 h, test compounds were added to the wells in a given concentration range. Then the cells were incubated for 72 h, after which a solution of MTT (3-(4,5-dimethyl-2-thiazolyl)-2,5-diphenyl-2H-tetrazolium bromide) was added to the cultures at 20 µL (5 mg/mL) to the well. After 2.5 h, the medium was removed from the wells and 200 µL of a mixture of DMSO/*i*-PrOH 1/1 was added. Optical density was measured on a plate spectrophotometer at a wavelength of 570 nm.

##### Statistical Analysis

Statistical data processing was carried out in the RStudio program (Version 1.4.1106 © 2022–2021 RStudio, PBC, Boston, MA, USA) using the R package (version 4.1.2). The cytotoxicity index (IC_50_) was calculated by plotting dose–response curves using the “drc” package [41].

## 4. Conclusions

Thus, in this work we extended the library of the 4,7-dihydro-6-nitroazolo[1,5-a]pyrimidine series, and also studied their antitumor properties. The inhibitory activity of these compounds against CK2 has been established, as well as their cytotoxic effect. Compounds of this series are comparable in inhibitory activity with the reference drug and exhibit a cytotoxic effect on tumor cells at micromolar concentrations. It is evident that the herein reported 4,7-dihydro-6-nitroazolo[1,5-a]pyrimidines have the potential to be studied as a new class of antitumor compounds.

## Data Availability

Data are contained within article.

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
