# Peer review of "CK2 Inhibition and Antitumor Activity of 4,7-Dihydro-6-nitroazolo[1,5-a]pyrimidines"

_molecules, 2022, doi:10.3390/molecules27165239_

Round 1
Reviewer 1 Report
The manuscript titled "CK2 inhibition and antitumor activity of 4,7-dihydro-6-nitroazolo[1,5-a]pyrimidines" by D.N. Lyapustin et al. presents the synthesis and biological studies of a series of pyrimidine derivatives. The syntheses are well described, and the identity of the compounds obtained was confirmed by NMR and IR spectra and elemental analysis.
The following are my comments on the manuscript.
Arguably, the heating rate when determining the melting point was misstated (25 degrees Celsius per minute).
Chemical shifts in NMR spectra in solution are given to two decimal places. This applies equally to proton and carbon spectra. The Authors reported chemical shifts for carbon atoms with one digit after the decimal point. For solid-state NMR spectra, chemical shifts are given to one decimal place. This is not applicable in this article.
The unit of wave number (sm-1 instead of cm-1) was incorrectly reported.
The experimental section on biological studies should be moved from the supplementary materials to the main body of the manuscript.
In Table 1, uncertainties usually have three significant digits (even four are given). Uncertainties cannot have more than two significant digits. In addition, the physical quantity must have the same decimal expansion as its uncertainty. Unfortunately, this is not met many times.
In the results and discussion section, it would be appropriate to expand the discussion on synthesis.
In Figures 1 and 2, the abbreviation "reference" should be terminated with a period, while if there are more references, the abbreviation "Refs" should be used.
Overall, the discussion of the results is quite narrow.
The supplementary material file takes a very long time to download.
Author Response
We would like to thank the reviewer for the work done. Please see the attachment

Reviewer 2 Report
The work on the “CK2 inhibition and antitumor activity of 4,7-dihydro-6-nitro-2 azolo[1,5-a] pyrimidines” is a valuable perfect and sutabile contribution to be published in Molecules Journal after justify some points.
Major points
· Firstly an extensive English revision is needed and typo errors were observed in the whole manuscript.
- Secondly, I would like to recommend the authors to re-write a lot of sentences due to the overlapping and high similarity rate with a lot of other articles especially in the matrials and method section
1- Abstract
· The abstract should be improved it seems just like a summery of results.
· Can you add a sentence regarding the chemical characterization methods of these compounds “ NMR, HRMS, IR “?? To abstract.
· Can you edit the line 16-17 and write positive control for the used molecule because the DMSO were used also in the evaluation as negative control.
· The Keywords should be edited no need to write nitro compounds then pyrimidines then nitrogen heterocycles, you can use one of them that will be enugh.
2- Introduction
· The introduction need improvemt as it is very short
· You can some statistical data regarding cancer to the introduction from recent publications, https://doi.org/10.1186/s13065-022-00839-5, as well as you can write a paragraph regarding the main biological targets in the treatment of cancer, DOI: 10.1055/a-0898-7347, because that were targted a spesicifc targets like CK-2 that will improve your story.
· Figure 1 and 2 should be edited, no need to write the ref. No on the structure, you can add them to the paragraph one by one for each structure.
· Line 47 edit these ref.s citation you have to crrect them as [30-32].
3- Results and discussion
· The chemistry section was well written
· Write the number of your intermiadtes and final compounds like aminoazoles (1 &2), aldehydes (4 ) ….etc
· You have to mention and write the main finding of NMR data, IR and discuss them.
· You have to identify the letters X, Y, R1 and R2 belongs to which groups in the scheme 1
· In table 1 no need to highlites the most potent compounds you can just make them as Bold font
· What do you mean of the values in minus, if there is no activity, and the inhibtion percentage are in minus values you should write no inhibtion better than – values in table 1.
· Edit the captions of table 1 and 2, they are unclear
· Line 85 edit limited to excluded because of their low ….
· Table 3 should have SD values
· Did you used positive control in the MTT assay ?? if yes add it s values to this table.
· I think you have to add a figure to show the selectivity ratio of your compounds against cancer cells Vs Normal cells, you can ref. this type of calculation form https://doi.org/10.1515/hc-2020-0134,
· The resolution of figure 3 should be improved accordingly.
4- Materials and Methods
· Regaring the NMR spectrum a lot of compounds has signals not signed and not belongs to your compounds, for example compound 5d has impurities, so can the authors use HPLC or HRMS to idnitify the percentage of these impurities.
· I think the MTT assay method should be in the main text not in the suppl. File.
5- The Conclusion was well written
6- Control again all references as the journal style
Best wishes
The work on the “CK2 inhibition and antitumor activity of 4,7-dihydro-6-nitro-2 azolo[1,5-a] pyrimidines” is a valuable perfect and sutabile contribution to be published in Molecules Journal after justify some points.
Major points
· Firstly an extensive English revision is needed and typo errors were observed in the whole manuscript.
- Secondly, I would like to recommend the authors to re-write a lot of sentences due to the overlapping and high similarity rate with a lot of other articles especially in the matrials and method section
1- Abstract
· The abstract should be improved it seems just like a summery of results.
· Can you add a sentence regarding the chemical characterization methods of these compounds “ NMR, HRMS, IR “?? To abstract.
· Can you edit the line 16-17 and write positive control for the used molecule because the DMSO were used also in the evaluation as negative control.
· The Keywords should be edited no need to write nitro compounds then pyrimidines then nitrogen heterocycles, you can use one of them that will be enugh.
2- Introduction
· The introduction need improvemt as it is very short
· You can some statistical data regarding cancer to the introduction from recent publications, https://doi.org/10.1186/s13065-022-00839-5, as well as you can write a paragraph regarding the main biological targets in the treatment of cancer, DOI: 10.1055/a-0898-7347, because that were targted a spesicifc targets like CK-2 that will improve your story.
· Figure 1 and 2 should be edited, no need to write the ref. No on the structure, you can add them to the paragraph one by one for each structure.
· Line 47 edit these ref.s citation you have to crrect them as [30-32].
3- Results and discussion
· The chemistry section was well written
· Write the number of your intermiadtes and final compounds like aminoazoles (1 &2), aldehydes (4 ) ….etc
· You have to mention and write the main finding of NMR data, IR and discuss them.
· You have to identify the letters X, Y, R1 and R2 belongs to which groups in the scheme 1
· In table 1 no need to highlites the most potent compounds you can just make them as Bold font
· What do you mean of the values in minus, if there is no activity, and the inhibtion percentage are in minus values you should write no inhibtion better than – values in table 1.
· Edit the captions of table 1 and 2, they are unclear
· Line 85 edit limited to excluded because of their low ….
· Table 3 should have SD values
· Did you used positive control in the MTT assay ?? if yes add it s values to this table.
· I think you have to add a figure to show the selectivity ratio of your compounds against cancer cells Vs Normal cells, you can ref. this type of calculation form https://doi.org/10.1515/hc-2020-0134,
· The resolution of figure 3 should be improved accordingly.
4- Materials and Methods
· Regaring the NMR spectrum a lot of compounds has signals not signed and not belongs to your compounds, for example compound 5d has impurities, so can the authors use HPLC or HRMS to idnitify the percentage of these impurities.
· I think the MTT assay method should be in the main text not in the suppl. File.
6- Control again all references as the journal style
Best wishes
Author Response

(The authors gave the same response as above.)

Reviewer 3 Report
The authors present CK2 inhibition and antitumor activity of 20 new pyrimidines. The structures were characterized by IR, 1H and 13C NMR and elemental analysis.
The manuscript is well written. The title, abstract, scheme, tables and figures of the manuscript are adequate to the content. The experimental part gives enough details about the experimental procedures. However the 13C NMR spectra are not assigned. The problem is the chemical shift of C7. In the article in JOC (ref. 30) in all compounds C7 resonate at 58-59 ppm. In this article C7 resonate at 38.2 ppm (5a), at 39.8 ppm (5c), even at 34.3 ppm (5d). The C7 resonance can be regarded as NMR check whether the heterocycle is formed. The included NMR spectra of compounds 5a-5d are spectra of compounds without pyrimidine ring. Please provide the NMR spectra of pure compounds 5a-5d or please reconsider their structure. Can you provide HSQC and HMBC spectra of the studied compounds? Additional remarks:Row 47 – square brackets are missing
Author Response

(The authors gave the same response as above.)

Reviewer 4 Report
This manuscript describes the synthesis and antitumor properties of new series of polysubstituted 4,7-dihydro-6-nitroazolo[1,5-a]pyrimidines. Some of the tested compounds are comparable in inhibitory activity with the reference drug and exhibit a cytotoxic effect on tumor cells at micromolar concentrations. The authors report encouraging data about some of the 4,7-dihydro-6-nitroazolo[1,5-a]pyrimidine derivatives with the potential to be studied as a new class of antitumor compounds. All of the compounds are well described and properly characterized by using different spectral measurements.
The manuscript provides a good amount of data and it is relevant to the journal. I recommend this manuscript for publication in Molecules.
Author Response
We would like to thank the reviewer for the work done
Round 2
Reviewer 1 Report
Although the Authors stated in their response that they had improved the notation of physical quantities along with their uncertainties, unfortunately this was not done. This applies to Tables 1 and 3. Uncertainties are given with up to two significant digits, and are always rounded up. After rounding the uncertainties, we round the physical quantity so that both numbers have the same decimal expansion.
Chemical shifts in NMR spectra in solution of both proton and carbon are given to two decimal places. The Authors reported the chemical shifts for carbon spectra with one digit after the decimal point. Why? In a previous review, I wrote that in spectra in solid-state the chemical shifts are given to one decimal place because of the poorer resolution than in solution spectra. Of course, this does not apply here, since the Authors did not record the spectrum for the solid only in solution.
The heating rate (25 degrees Celsius per minute) in determining the melting point is much too high. It is therefore not surprising that the Authors give a melting point range of a dozen degrees. For pure substances, the melting point range should not exceed 1 degree. When the temperature is 20 degrees lower than the expected melting point, the heating rate should be reduced to a maximum of two degrees per minute. Please repeat the melting point measurements, as the data presented are not reliable.
Reviewer 2 Report
almost all requested documents were justified, just i am still believed that the overlapping in the showed be improved by change the phrases of the method section.
positive control like well known anticancer drug for example Doxorubicin can be added to the experiments to compare your results
